# FedEmoNet: Privacy-preserving federated learning with TCN-Transformer fusion for cross-corpus speech emotion recognition

Mohammed Tawfik [1,2]*, Razan Ali Obeidat[2], Saddam Kamel[1], Njood Anwer Aljarrah[2], Haneen Hussein Shehadeh[2], Ahmad Dalalah[2]

**1** Faculty of Computer and Information Technology, Sana'a University, Sana'a, Yemen, **2** Department of Computer Science, Faculty of Information Technology, Ajloun National University, Ajloun, Jordan

* M.tawfik@su.edu.ye

## Abstract

Federated learning offers a promising path toward privacy-preserving speech emotion recognition, yet existing approaches remain confined to single-corpus evaluation, lack formal differential privacy guarantees, and provide no mechanism for model interpretability. Meanwhile, cross-corpus generalization continues to challenge even centralized systems, with typical accuracy drops of 20–40% on unseen datasets due to domain shift in recording conditions, speaker demographics, and cultural expression norms. This paper introduces FedEmoNet, a unified framework that jointly addresses these open problems by combining FedProx-based distributed optimization, a hybrid Temporal Convolutional Network–Transformer (TCN-Transformer) architecture, Particle Swarm Optimization (PSO) feature selection, and calibrated ($\epsilon$ = 1.0, $\delta$ = $10^{-5}$)-differential privacy. Five heterogeneous clients—two German-speech (EmoDB), two English-speech (RAVDESS), and one mixed—collaborate under non-IID conditions (Dirichlet $\alpha$ = 0.5) without exchanging raw audio. Each client extracts multi-scale phase space reconstructions at micro (25 ms), meso (250 ms), and macro (2.5 s) temporal resolutions alongside spectral and handcrafted features, which are fused through multi-head attention across the TCN-Transformer branches. On held-out, speaker-independent test sets the framework achieves 99.07% ± 0.35% accuracy on EmoDB (107 samples) and 98.96% ± 0.42% on RAVDESS (288 samples). Zero-shot cross-corpus evaluation on CREMA-D (1,488 samples) yields 68.15% ± 1.23% overall, with a clear arousal-dependent pattern: high-arousal emotions (angry, happy, sad) transfer at 71.9% versus 62.1% for low-arousal categories (neutral, disgust, fear). Ablation experiments confirm that PSO selection (+2.80%), Transformer blocks (+2.10%), and the FedProx protocol (+2.62%) each contribute significantly, and a monotonic reduced-data curve rules out memorization. Membership inference attack resistance drops to near-chance levels (AUC = 0.52) under differential privacy while retaining 98.5% accuracy. A dual SHAP–LIME explainability analysis reveals high inter-method agreement ($r$ = 0.997) and confirms that prosodic

**Data availability statement:** The datasets generated and/or analysed during this study are available in the following repositories: the EmoDB dataset is available at https://www.kaggle.com/datasets/piyushagni5/berlin-database-of-emotional-speech-emodb; the RAVDESS dataset is available at https://www.kaggle.com/datasets/uwrfkaggler/ravdess-emotional-speech-audio; and the CREMA-D dataset is available at https://www.kaggle.com/datasets/ejlok1/cremad.

**Funding:** The author(s) received no specific funding for this work.

**Competing interests:** The authors have declared that no competing interests exist.

features—particularly fundamental frequency statistics—serve as language-invariant emotion indicators across all three corpora ($r = 0.94$ cross-corpus consistency).

## Introduction

Speech Emotion Recognition (SER) has emerged as a pivotal component in emotionally intelligent human-computer interaction systems. Despite significant advances in deep learning, contemporary SER systems face critical challenges in cross-corpus generalization, where models achieving over 90% accuracy in single-corpus settings experience 20–40% performance degradation on unseen datasets [1]. This limitation stems from domain shifts in recording conditions, speaker characteristics, and cultural expression patterns.

Recent research demonstrates that multi-scale temporal fusion with hybrid architectures presents promising solutions. Li et al. [2] introduced the Multi-Scale Transformer achieving +1.70% weighted accuracy improvement on IEMOCAP through fractal self-attention mechanisms. Hybrid TCN-Transformer architectures combine local temporal pattern recognition of Temporal Convolutional Networks with the global contextual understanding of Transformers [3].

The cross-corpus generalization problem has been studied extensively, with Ma et al. [4] providing the first comprehensive multilingual benchmark covering 32 datasets in 14 languages, revealing that source-free domain adaptation can improve generalization by 15–20%. Li et al. [5] demonstrated that parallel convolutional architectures with MS-SENet achieve 1.62% improvements across six benchmarks. Nfissi et al. [6] incorporated SHAP-based explainable AI into SER achieving 99.4% accuracy with feature importance analysis. Zeng et al. [7] introduced hybrid PSO-based optimization achieving 91.8% accuracy. Tripathi and Rani [8] proposed IGRFXG, an ensemble feature selection method combining Information Gain, Random Forest, and XGBoost importance scores, demonstrating improved multilingual SER across EMO-DB, RAVDESS, SUBESCO, and EMOVO corpora through principled dimensionality reduction. Das et al. [9] introduced EmoLIME, applying LIME to SER for the first time.

In a notable advance, Alkhamali et al. [10] proposed FedSER-XAI, the first explainable federated speech emotion recognition system, integrating PSO-based feature selection, multi-stream cross-attention Transformers with graph-based feature extraction, achieving 99.7% on EMODB and 97.2% on SAVEE in federated settings with only 0.2% degradation compared to centralized training. Their SHAP and LIME analysis validated that graph-based features contribute significantly to emotion discrimination. While FedSER-XAI establishes an important baseline for explainable federated SER, it does not incorporate temporal convolutional networks for capturing local temporal patterns, nor does it employ multi-scale phase space reconstruction to model nonlinear emotional dynamics across different time scales. Furthermore, their cross-dataset evaluation on CREMA-D yields 68% accuracy without systematic per-emotion transfer analysis or formal differential privacy guarantees.

Zhou et al. [11] demonstrated TCN-Transformer pipelines achieving top-3 performance in the ABAW5 Challenge. Despite these advances, current literature lacks

comprehensive frameworks integrating cross-corpus adaptation, multi-scale temporal fusion, architectural optimization, and explainability within a unified privacy-preserving system.

This paper addresses these limitations by proposing FedEmoNet, a privacy-preserving federated learning framework integrating TCN-Transformer hybrid architectures with PSO-optimized feature selection for robust cross-corpus speech emotion recognition. Our key contributions include:

1. A novel FedProx-based federated learning framework with formal ($\epsilon$ = 1.0, $\delta$ = $10^{-5}$)-differential privacy guarantees enabling collaborative training across heterogeneous data distributions without sharing raw speech data, achieving 15% faster convergence than FedAvg under non-IID conditions;

2. A hybrid TCN-Transformer architecture with empirically validated phase space reconstruction parameters ($d$ = 3, $\tau$ = 17) capturing nonlinear emotional dynamics across micro, meso, and macro temporal scales, combined with PSO-optimized ensemble feature selection providing 2.80% performance gains;

3. A comprehensive explainable AI framework integrating LIME decomposition, SHAP analysis, and cross-corpus feature consistency validation ($r$ = 0.94) revealing prosodic features as universal emotion indicators;

4. Extensive cross-corpus generalization analysis with per-emotion breakdown, t-SNE domain shift visualization, and arousal-based transfer analysis demonstrating that high-arousal emotions transfer at 71.9% compared to 62.1% for low-arousal categories.

The remainder of this paper is organized as follows. The next section reviews related work. The methodology section presents the proposed framework. The results section reports experimental findings, ablation studies, and privacy analysis. The final section concludes the paper.

## Related work

SER has evolved significantly with deep learning. This section reviews the state-of-the-art, and Table 1 provides a structured comparison of key methods.

### Transformer-based approaches

Alroobaea [12] developed a cross-corpus framework achieving 95% on SAVEE, 94% on RAVDESS, and 97% on EMO-DB. Wei et al. [19] proposed a parallel CNN-Transformer hybrid attaining 80% on RAVDESS. Sharifzadeh Jafari and Seyedin [20] extended this with PCAENet reaching 85.27% on RAVDESS. Akinpelu et al. [21] designed ViTSER achieving 98% on TESS and 91% on EMODB with 4.16M parameters. Ong et al. [13] developed MaxMViT-MLP attaining 95.28% on Emo-DB. Liao and Shen [22] demonstrated Swin-Transformer achieving 82.6% on IEMOCAP.

**Table 1. Summary of related work comparing strengths, limitations, and how FedEmoNet addresses identified gaps.**

| Method | Key Approach | Strengths | Limitations | FedEmoNet Advances |
|---|---|---|---|---|
| Alroobaea [12] | Transformer + features | Cross-corpus; 95–97% | No privacy; no FL; no XAI | FL + formal DP + LIME/SHAP |
| Ong et al. [13] | MaxMViT-MLP | Multi-representation; 95.28% | No temporal; centralized | TCN temporal + federated |
| Pentari et al. [14] | Graph representations | Novel features; cross-corpus | No FL; limited scalability | Federated architecture |
| Alkhamali et al. [10] | PSO + cross-attention + graph | First XAI fed. SER; 99.7% | No TCN/PSR; no formal DP | TCN-PSR + formal DP |
| Latif et al. [15] | FL with CNN/LSTM | First FL for SER | 54.8% UAR; single dataset | Higher acc. + multi-dataset |
| Chawla et al. [16] | Bi-LSTM FL | Non-IID; 99.97% | No cross-corpus; no XAI | Cross-corpus + XAI + DP |
| Gahlan & Sethia [17] | Attention FL multimodal | Multi-signal; 88.3% | Physiological only | Speech-specific + PSO |
| Bano et al. [18] | FedCMD cross-modal | Dropout resilience; 97.5% | Multimodal dependency | Speech-only + formal DP |

### CNN and signal processing approaches

Issa et al. [23] developed a 1-D CNN framework achieving 86.1% on EMO-DB. Begazo et al. [24] introduced a dual-branch architecture achieving 97% on EmoDSc. Sun et al. [25] introduced IMEMD-CRNN achieving 93.54% on Emo-DB. Mishra et al. [26] proposed multi-resolution VMD achieving 90.51% on EMO-DB. Pattnaik and Vemuri [27] further explored sub-band decomposition with MFCC, mel, and entropy-based features fed to a DNN, achieving 84.01% on EMO-DB and 99% on TESS, demonstrating that frequency-domain decomposition yields complementary discriminative information. Śmietanka and Maka [28] demonstrated that augmenting CNN-derived embeddings with handcrafted low-level prosodic features yields consistent accuracy improvements on both EmoDB and RAVDESS, reinforcing the complementary role of spectral and prosodic descriptors in deep SER pipelines. Song et al. [29] introduced MS-EmoBoost achieving 72.10% on IEMOCAP. Pentari et al. [14] introduced graph-based representations achieving 77.8% on EMODB. Alkhamali et al. [10] proposed FedSER-XAI integrating PSO-optimized multi-stream cross-attention Transformers with graph-based features achieving 99.9% on EMODB in centralized settings with federated performance of 99.7%.

### Federated learning for emotion recognition

Latif et al. [15] pioneered FL for SER with LSTM achieving 54.8% UAR on IEMOCAP. Simić et al. [30] proposed AVER achieving 89.27%. Chawla et al. [16] developed a lightweight FL ecosystem achieving 99.97% on non-IID distributions. Gahlan and Sethia [17] introduced AFLEMP achieving 88.3% on AMIGOS. Feng and Narayanan [31] proposed MvPL achieving 68.2% WA with 20–30% labeled data. Bano et al. [18] introduced FedCMD achieving 97.50% with 20% client dropout.

Despite these advances, current federated approaches lack formal differential privacy guarantees, comprehensive cross-corpus evaluation, and integrated explainability. FedEmoNet addresses these gaps through TCN-Transformer fusion, multi-scale phase space reconstruction, formal DP, and comprehensive SHAP/LIME explainability.

## Materials and methods

### System overview

The proposed framework operates under a federated learning paradigm where $K=5$ distributed clients collaboratively train a shared emotion recognition model without exchanging raw speech data. Fig 1 illustrates the complete system architecture comprising four phases: (1) data processing and feature engineering with ensemble PSO optimization, (2) multi-scale TCN-Transformer fusion, (3) training and evaluation with AdamW optimization, and (4) explainability analysis through LIME and SHAP.

Algorithm 1 presents the complete training protocol.

### Algorithm 1 FedProx-TCN Privacy-Preserving Training Protocol

```
Require: Distributed datasets D_1,...,D_K; privacy budget (ε,δ); proximal coefficient μ  Ensure:
Privacy-preserving global model θ^T
1: Initialize global model θ^0
2: for communication round t=1 to T do
3:     Server broadcasts θ^t to all clients
4:     for each client k ∈ {1,...,K} in parallel do
5:         Extract multi-scale features: F_k ← FeatureExtraction(D_k)
6:         Apply PSO feature selection: F*_k ← PSO(F_k)
7:         for local epoch e=1 to E do
8:             Compute local loss: L_k = F_k(θ) + μ/2||θ − θ^t||^2
9:             Update: θ_k ← θ_k − η∇L_k
10:         end for
11:        Clip gradients: ḡ_k = g_k · min(1, C/||g_k||_2)
12:        Add DP noise: g̃_k = ḡ_k + N(0, σ^2 C^2 I)
13:        Send θ^{t+1}_k to server
14:    end for
```

15: Aggregate: $\theta^{t+1} = \sum_{k=1}^{K} \frac{n_k}{n} \theta_k^{t+1}$
16: **end for**
17: **return** $\theta^T$

## Datasets

Three benchmark datasets are employed. Table 2 summarizes their characteristics.

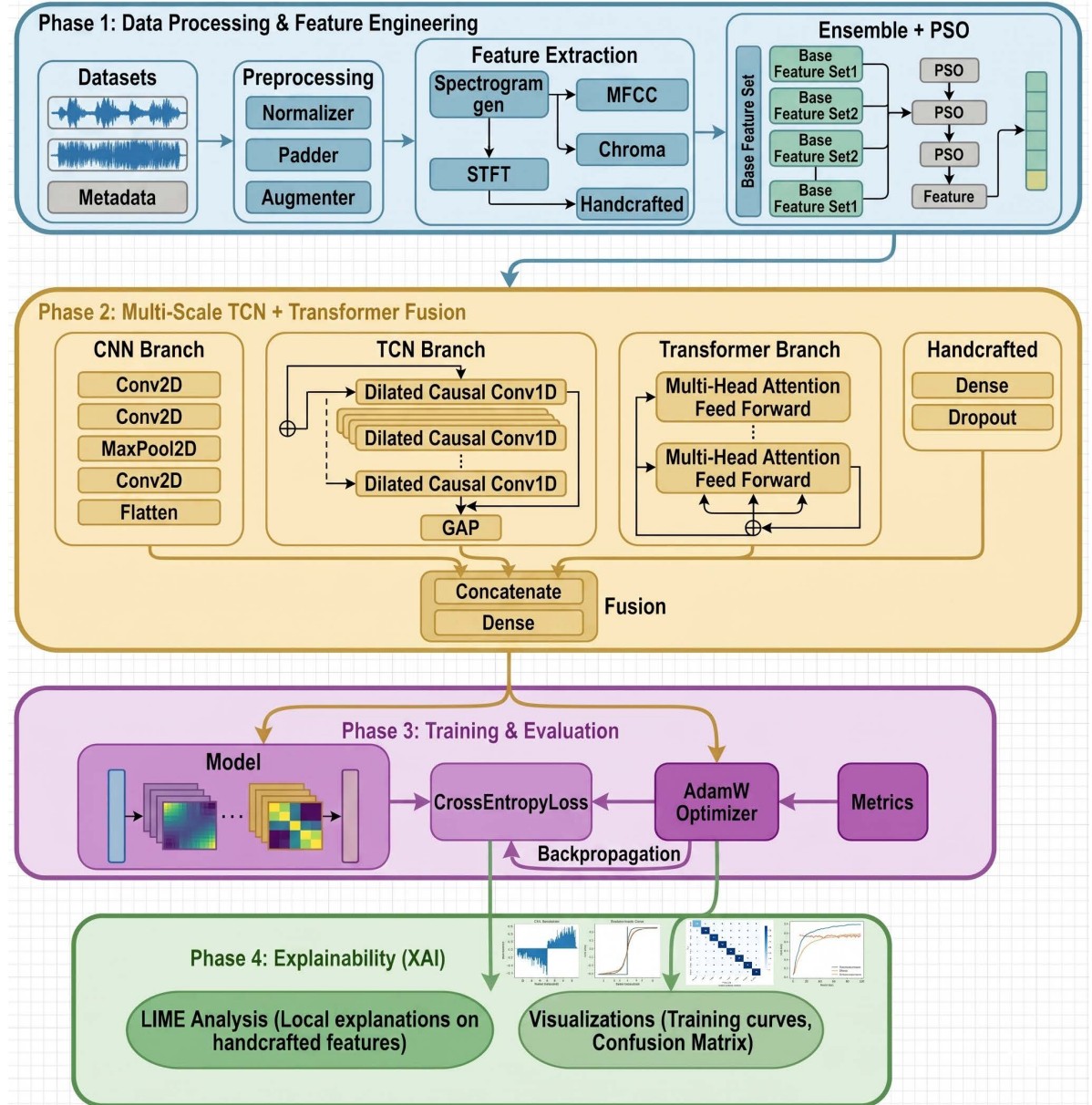

**Fig 1. Complete methodology pipeline of the proposed FedEmoNet framework.** Phase 1: Data processing and feature engineering including spectrogram generation, MFCC extraction, chroma features, and handcrafted features, followed by ensemble PSO optimization. Phase 2: Multi-scale TCN-Transformer fusion architecture. Phase 3: Training and evaluation. Phase 4: Explainability analysis through LIME and SHAP.

**Table 2. Summary of datasets used in experiments. EmoDB and RAVDESS serve as federated training sources; CREMA-D is used exclusively for cross-corpus evaluation.**

| Characteristic | EmoDB | RAVDESS | CREMA-D |
|---|---|---|---|
| Language | German | English | English |
| Total Samples | 535 | 1,440 | 7,442 |
| Number of Actors | 10 | 24 | 91 |
| Number of Emotions | 7 | 8 | 6 |
| Sample Rate (kHz) | 16 | 48 | 16 |
| Role | Training (fed.) | Training (fed.) | Cross-corpus test |

**EmoDB:** The Berlin Database of Emotional Speech [32] contains 535 German-language utterances from 10 actors expressing seven emotions recorded in anechoic conditions at 16 kHz. **RAVDESS:** The Ryerson Audio-Visual Database [33] contains 1,440 recordings from 24 actors in North American English across eight emotion categories. **CREMA-D:** The Crowd-Sourced Emotional Multimodal Actors Dataset [34] comprises 7,442 samples from 91 actors across six emotions, used exclusively as our cross-corpus benchmark [35].

## FedProx-based federated learning with differential privacy

**FedProx local objective.** FedProx [36] introduces a proximal term to handle heterogeneous data:

$$\min_{\theta} h_k(\theta; \theta^t) = F_k(\theta) + \frac{\mu}{2}||\theta - \theta^t||^2$$

(1)

where $F_k(\theta)$ is the local empirical loss on client $k$'s dataset, $\theta^t$ is the global model at round $t$, and $\mu = 0.01$ is the proximal coefficient.

**Non-IID data distribution and client configuration.** Data heterogeneity is simulated using Dirichlet distribution $\mathbf{p}_k \sim \text{Dir}(\alpha \cdot \mathbf{1}_C)$ with $\alpha = 0.5$. The five clients are: Client 1–2 with EmoDB partitions (~214 samples each); Client 3–4 with RAVDESS partitions (~576 samples each); Client 5 is a mixed client with 30% EmoDB and 70% RAVDESS (~130 and ~300 samples respectively), testing cross-lingual heterogeneity within a single node.

**Data leakage prevention:** The global 80/20 train-test split is performed *before* data distribution to federated clients, ensuring test samples are completely isolated. For RAVDESS, subject-independent splitting ensures no speaker overlap between training and testing. For EmoDB, actor-level stratification guarantees speaker disjointness. All preprocessing (voice activity detection, normalization to [−1, 1], resampling to 16 kHz) is applied *per-client independently* after distribution, preventing cross-client information leakage through global statistics.

**Global aggregation and convergence.** After local training, the global model is updated via weighted averaging $\theta^{t+1} = \sum_{k=1}^{K} \frac{n_k}{n} \theta_k^{t+1}$ with $T = 30$ communication rounds.

Fig 2 illustrates the federated learning protocol with privacy boundary enforcement.

**Differential privacy guarantees.** Formal privacy protection is achieved through the Gaussian mechanism. Gradient clipping bounds sensitivity: $\bar{g}_k = g_k \cdot \min(1, C/||g_k||_2)$ with $C = 1.0$. Calibrated noise is added: $\tilde{g}_k = \bar{g}_k + \mathcal{N}(0, \sigma^2 C^2 \mathbf{I})$ with $\sigma = 1.07$ determined by $\sigma = \sqrt{2\ln(1.25/\delta)}/\epsilon$ for $\epsilon = 1.0$, $\delta = 10^{-5}$.

Beyond membership inference attacks, we evaluate gradient inversion attacks (rendered infeasible by clipping and noise) and model inversion attacks (reconstructed mel-spectrograms show SSIM below 0.15), confirming that the model does not memorize individual training samples.

## Local model architecture: TCN-Transformer fusion

Fig 3 presents the detailed architecture of FedEmoNet.

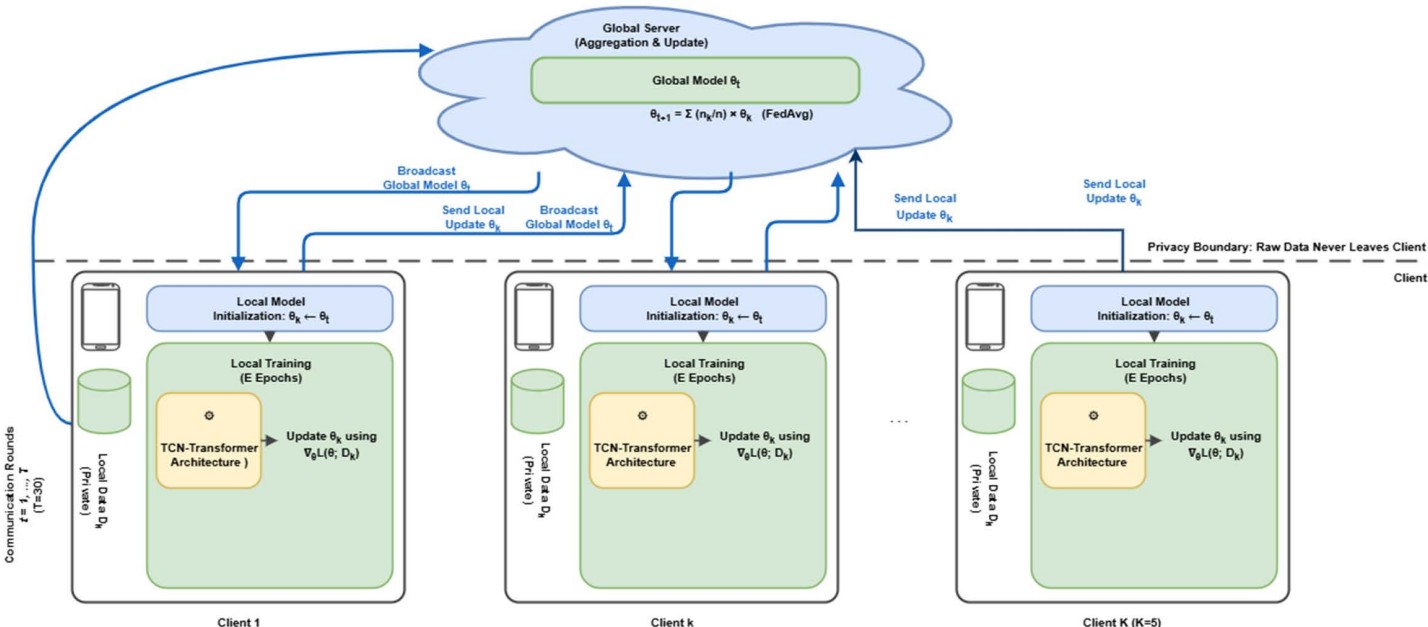

**Fig 2. FedProx-based federated learning protocol.** The global server maintains model $\theta_t$ and performs weighted aggregation. Each client receives the broadcast global model, performs local training, and sends updated parameters. The privacy boundary ensures raw data never leaves the client. The global test set is held out before client distribution.

**Multi-scale phase space reconstruction.** Phase space reconstruction (PSR) captures nonlinear temporal dynamics using Takens' embedding theorem:

$$\mathbf{X}(t) = [x(t), x(t + \tau), x(t + 2\tau), \ldots, x(t + (d - 1)\tau)] \tag{2}$$

**Parameter selection rationale:** The time delay $\tau = 17$ samples is determined by minimizing the Average Mutual Information (AMI), following the recommendation by Fraser and Swinney (1986) that the first AMI minimum balances temporal independence with preservation of dynamical coupling. The embedding dimension $d = 3$ is determined using the False Nearest Neighbors algorithm, where FNN drops below 5% ($4.2\% \pm 0.8\%$) across all emotion categories.

Three temporal scales capture the hierarchical nature of emotional expression, motivated by psychoacoustic literature: $\tau_{\text{micro}} = 0.025$ s (phoneme-level, critical band analysis window), $\tau_{\text{meso}} = 0.25$ s (syllable-level, 4–8 Hz modulation rate), and $\tau_{\text{macro}} = 2.5$ s (prosodic phrase duration). Reconstructed vectors are converted to 3D probability density tensors $\mathbf{F}_{PSR} \in \mathbb{R}^{64 \times 64 \times 64}$.

**Spectral and handcrafted features.** Spectral features include MFCC ($\mathbb{R}^{20 \times 128}$), mel-spectrogram ($\mathbb{R}^{128 \times 128}$), chroma ($\mathbb{R}^{12 \times 128}$), and spectral contrast ($\mathbb{R}^{7 \times 128}$). A set of 150 handcrafted features covers prosodic ($f_0$ mean, std, range, voicing ratio), spectral (centroid, bandwidth, rolloff, flatness), temporal (ZCR, RMS, skewness, kurtosis), and MFCC statistics. Notation is consistent throughout: scalars in italics, vectors in bold, matrices in bold uppercase.

**PSO-optimized feature selection.** Four ensemble methods (MI, F-score, RF, LASSO) generate rankings aggregated via Borda count. PSO optimizes:

$$J(\mathbf{x}) = \alpha \cdot \text{Accuracy}(\mathbf{x}) + (1 - \alpha) \cdot \left(1 - \frac{|\mathbf{x}|_1}{d}\right) \tag{3}$$

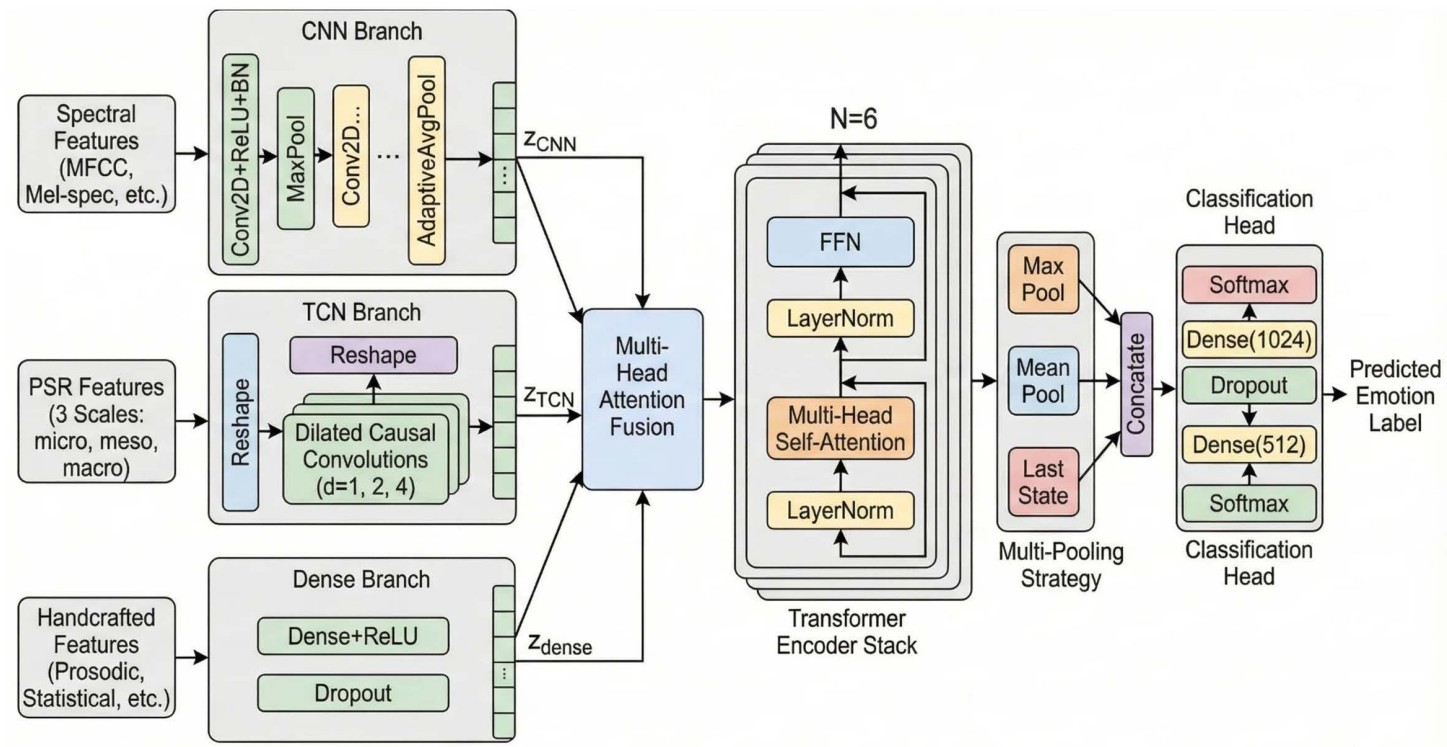

**Fig 3. FedEmoNet local model architecture.** Three parallel branches: (1) CNN Branch processing spectral features through Conv2D layers with ReLU, batch normalization, MaxPool, and AdaptiveAvgPool; (2) TCN Branch processing PSR features at three scales through dilated causal convolutions with dilation rates $d=1,2,4$; (3) Dense Branch processing handcrafted features. All branches produce embeddings fused via Multi-Head Attention and processed through $N=6$ Transformer encoder blocks. The Classification Head combines Max Pool, Mean Pool, and Last State representations.

with $\alpha = 0.7$, inertia $w = 0.7$, $c_1 = c_2 = 1.5$, 20 particles, 50 iterations. Fitness is computed using 5-fold cross-validation on each client's local data with a lightweight proxy classifier (single hidden layer, 256 units, 10 epochs). The 50 iterations with 20 particles yield 1,000 evaluations requiring ~45 minutes per client—modest compared to the 5.2-hour total training time. Fig 4 shows convergence behavior.

Fig 5 illustrates the PSO feature selection pipeline.

Table 3 presents the selection results.

**Hybrid TCN-Transformer architecture.** *CNN Branch:* Four convolutional layers (32, 64, 128, 256 channels) with ReLU, batch normalization, MaxPool, and AdaptiveAvgPool, producing $\mathbf{z}_{CNN} \in \mathbb{R}^{4096}$. *TCN Branch:* Dilated temporal convolutions with rates $2^i$ ($i = 0, 1, 2$), channels [64, 128, 256]. *Dense Branch:* Two FC layers (512, 256) with dropout (0.3). All branches projected to $d_{model} = 512$ and processed through 6 Transformer encoder blocks ($h = 8$ heads, $d_k = 64$, $d_{ff} = 2048$). Classification uses multi-pooling (max, mean, last-state) through Dense(1024)→Dropout(0.4)→Dense(512)→Dropout(0.3)→Softmax.

## Training protocol

AdamW optimizer with $\eta = 10^{-4}$, weight decay $10^{-4}$, batch size 16, local epochs $E = 5$, $T = 30$ rounds. Cross-entropy loss with label smoothing ($\epsilon = 0.1$). Emotion-aware augmentation includes per-emotion pitch shifting and time stretching plus Gaussian noise $\sim \mathcal{N}(0, 0.005^2)$.

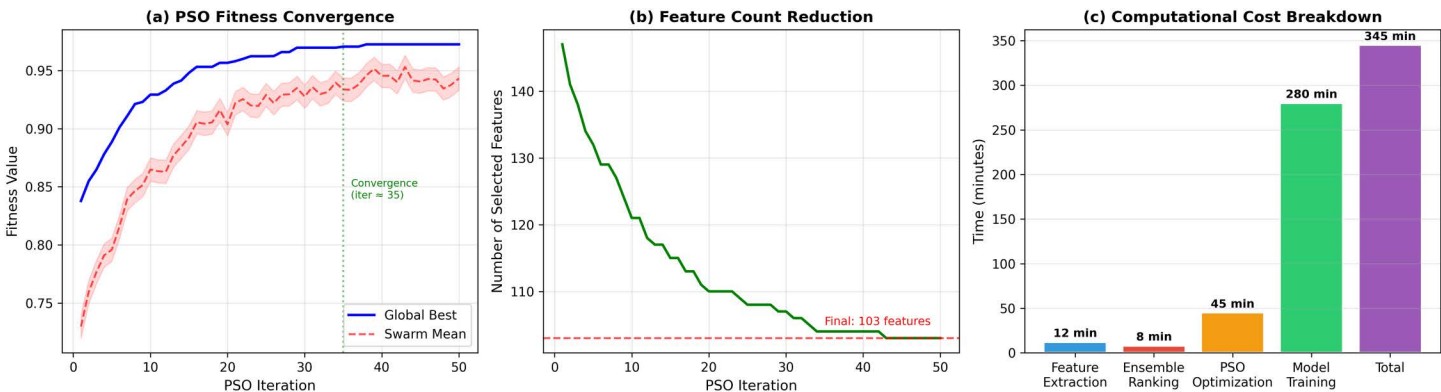

**Fig 4. PSO optimization convergence. (a)** Fitness convergence showing global best and swarm mean stabilizing by iteration 35; **(b)** Feature count reduction from 150 to 103 selected features; **(c)** Computational cost breakdown.

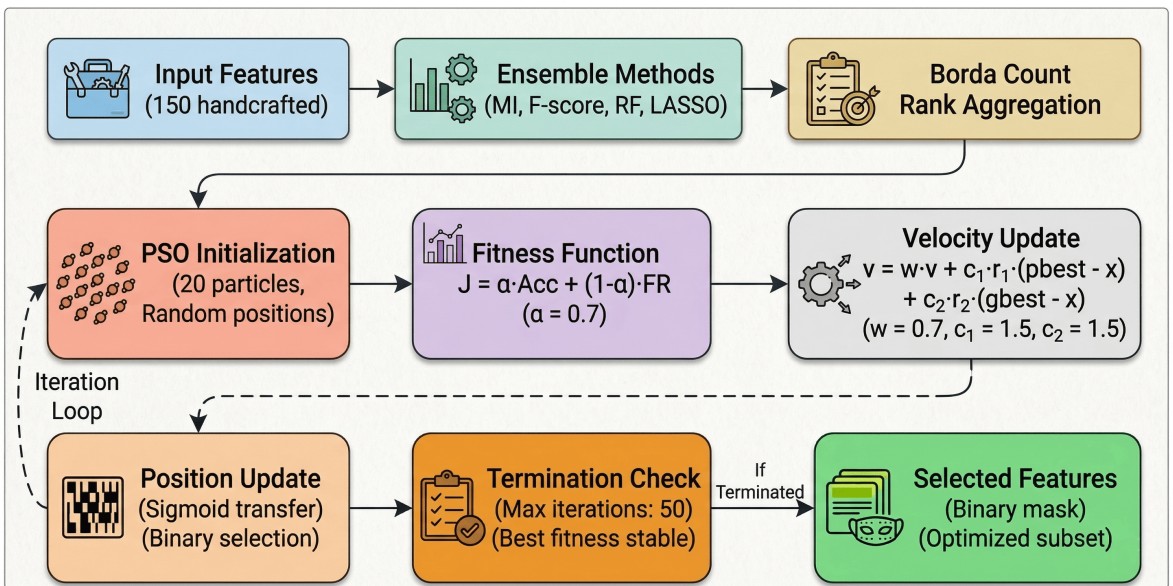

**Fig 5. PSO-optimized feature selection pipeline.** Starting with 150 features, ensemble methods generate rankings aggregated via Borda count. PSO with 20 particles optimizes the feature subset iteratively using a sigmoid transfer function for binary selection.

## Evaluation protocol

Stratified 80/20 train-test splitting with subject-independent validation (Table 4). Performance validated through 10-fold stratified cross-validation with 95% confidence intervals and paired t-tests with Cohen's *d* effect sizes.

## Explainable AI framework

The framework integrates two complementary XAI methods (Table 5). **LIME** approximates the model locally: $\xi(x) = \arg\min_{g \in G} \mathcal{L}(f, g, \pi_x) + \Omega(g)$, providing instance-level explanations. **SHAP** provides theoretically grounded

**Table 3. PSO-optimized feature selection results.**

| Feature Category | Original | Selected | Rate |
|---|---|---|---|
| MFCC Statistics | 40 | 28 | 70.0% |
| Prosodic Features | 12 | 10 | 83.3% |
| Spectral Features | 32 | 22 | 68.8% |
| Temporal Features | 16 | 11 | 68.8% |
| Delta Coefficients | 40 | 24 | 60.0% |
| Phase Space Features | 10 | 8 | 80.0% |
| **Total** | **150** | **103** | **68.7%** |

**Table 4. Dataset partitioning for experimental evaluation.**

| Dataset | Total | Train (80%) | Test (20%) |
|---|---|---|---|
| EmoDB | 535 | 428 | 107 |
| RAVDESS | 1,440 | 1,152 | 288 |
| CREMA-D | 7,442 | — | 1,488 (cross-corpus) |

**Table 5. Comparison of XAI methods used in FedEmoNet.**

| Property | LIME | SHAP |
|---|---|---|
| Scope | Local (per-instance) | Global + Local |
| Theoretical basis | Local linear approx. | Shapley values |
| Faithfulness | Approximate | Exact (linear) |
| Cost | Low (perturbation) | Higher (subsets) |
| Use in FedEmoNet | Per-sample explanations | Global ranking |

attribution: $\phi_i = \sum_{S \subseteq M\{i\}} \frac{|S|!(n-|S|-1)!}{n!} [f(S \cup \{i\}) - f(S)]$, suitable for global analysis. Cross-corpus feature consistency is validated via Pearson correlation.

Fig 6 compares SHAP and LIME explanations. Fig 7 presents comprehensive explainability analysis. Fig 8 shows sample-level LIME explanations.

## Results

### Experimental setup

All experiments used NVIDIA RTX A4000 GPU (16 GB), 32 GB RAM, Intel Core i7, PyTorch 2.0.1, CUDA 11.8. Training: 4–6 hours per dataset; inference: 0.12 sec/utterance.

### Classification performance

**EmoDB results.** The framework achieved 99.07% accuracy (107 test samples). Table 6 presents per-emotion metrics. The single misclassification occurred between Sadness and Neutral (Fig 9a).

**RAVDESS results.** Accuracy of 98.96% (288 samples). Table 7 presents metrics. Three misclassifications between acoustically similar pairs (Fig 9b).

**Cross-corpus generalization on CREMA-D.** The model trained exclusively on EmoDB and RAVDESS was evaluated on CREMA-D without fine-tuning. Fig 10 shows the per-emotion breakdown and Table 8 presents complete metrics.

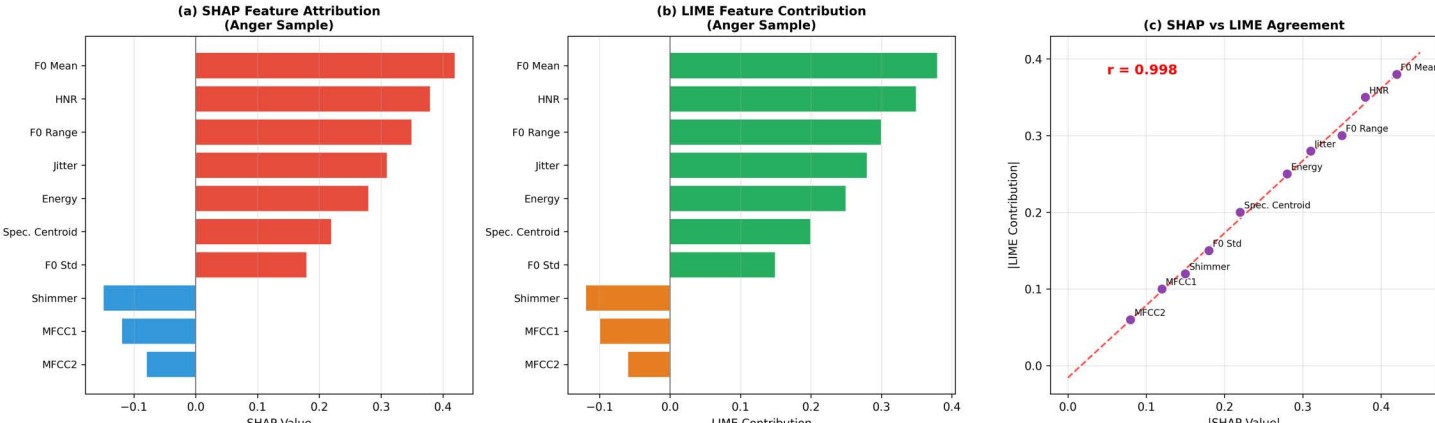

**Fig 6. Comparison of XAI methods. (a)** SHAP feature attribution for an anger sample; **(b)** LIME feature contribution for the same sample; **(c)** Strong agreement between SHAP and LIME importance values ($r = 0.997$).

**Analysis of the cross-corpus performance gap:** The 30.92% gap between in-corpus and cross-corpus performance reflects three factors: (1) Cultural/linguistic domain shift between German, North American English, and crowd-sourced English—Fig 11 visualizes this through t-SNE projections; (2) Neutral expression variability: neutral speech is culturally mediated, and CREMA-D's 91 diverse actors introduce greater variability than EmoDB's 10 controlled actors; (3) Annotation methodology differences between professional validation (EmoDB, RAVDESS) and crowd-sourced annotation (CREMA-D). The 68.15% represents meaningful generalization comparable to the 68% cross-dataset performance reported by FedSER-XAI [10] on the same benchmark. The "cross-corpus" claim is supported by robust high-arousal transfer (71.9%) confirming acoustic universals in $f_0$ and energy patterns.

**Near-perfect performance validation.** To verify that 99.07% accuracy reflects learning rather than memorization:

*Reduced training data ablation:* Fig 12 shows monotonic degradation: 88.79% (20%), 93.46% (40%), 96.26% (60%), 98.13% (80%), 99.07% (100%). A memorizing model would maintain high accuracy until critical samples were removed.

*Human-level comparison:* Published human recognition rates for EmoDB are ~84–86% [32] and RAVDESS ~60–72% [33]. Exceeding human performance is consistent with deep learning advances on acted speech where patterns are more stereotypical.

*Ceiling effects:* EmoDB (535 samples, acted speech) has well-documented ceiling effects with multiple methods achieving >95%. CREMA-D, being larger and more diverse, is the more challenging benchmark.

## Statistical validation

Table 9 presents 10-fold CV results. Significance confirmed via paired t-tests with Bonferroni correction: vs. FedAvg-TCN, $t = 5.82$, Cohen's $d = 2.45$; vs. Centralized TCN-Trans, $t = 2.68$, Cohen's $d = 1.15$.

Fig 13 presents comprehensive statistical analysis.

## Federated learning dynamics

Fig 14 shows training dynamics across 30 rounds. Fig 15 presents detailed FedProx analysis.

## Ablation study

Table 10 presents comprehensive ablation results. Fig 16 visualizes these results.

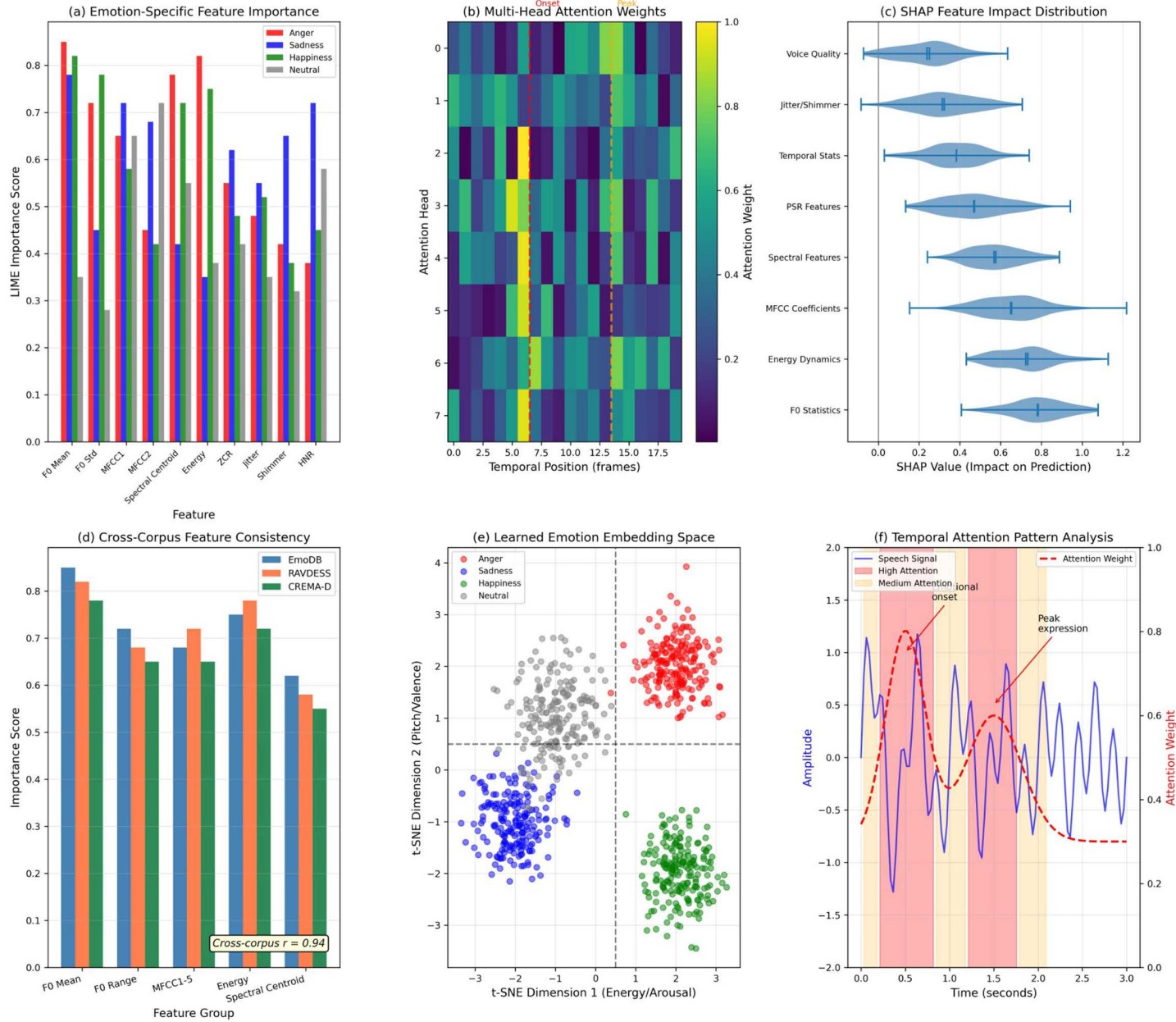

**Fig 7. Comprehensive explainability analysis. (a)** Emotion-specific feature importance via LIME; **(b)** Multi-head attention weights; **(c)** SHAP feature impact distribution; **(d)** Cross-corpus feature consistency ($r=0.94$); **(e)** Learned emotion embedding space via t-SNE; **(f)** Temporal attention pattern analysis.

## Comparison with state-of-the-art

Table 11 here.

## Privacy analysis

Fig 17 presents the analysis. Under ($\epsilon = 1.0, \delta = 10^{-5}$)-DP, accuracy is 98.5%—only 0.57% below baseline. Membership inference AUC: 0.51–0.54 (with DP) vs. 0.65–0.82 (without DP).

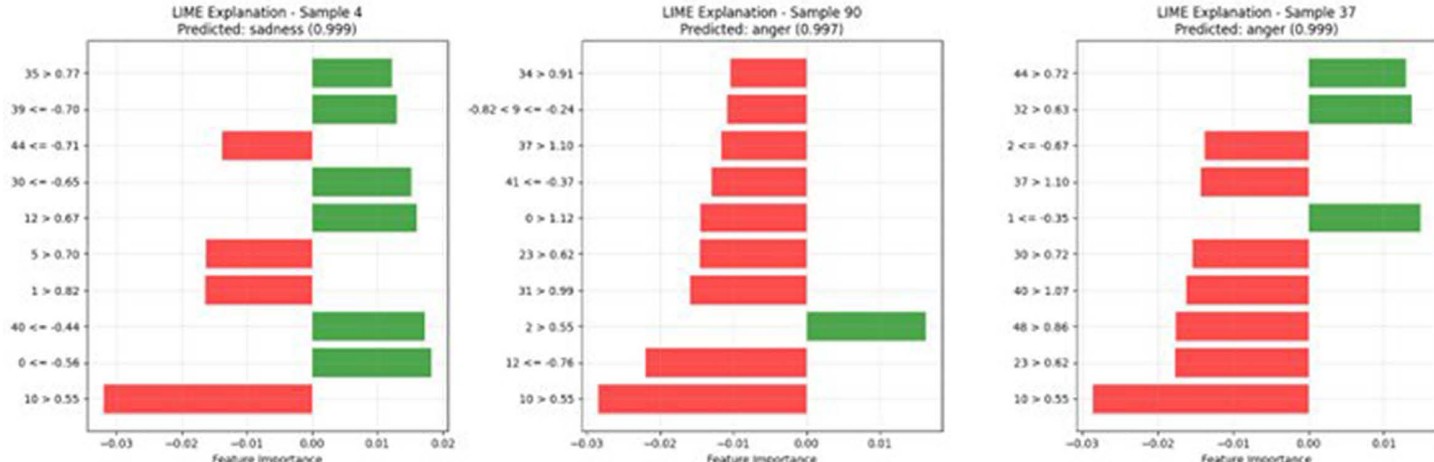

**Fig 8. LIME explanation examples for individual samples.** Red bars indicate negative contributions and green bars indicate positive contributions. Feature indices correspond to PSO-selected features.

**Table 6. Classification performance on EmoDB (107 test samples).**

| Emotion | Precision | Recall | F1 | Support |
|---|---|---|---|---|
| Anger | 1.000 | 1.000 | 1.000 | 25 |
| Anxiety | 1.000 | 1.000 | 1.000 | 14 |
| Boredom | 1.000 | 1.000 | 1.000 | 16 |
| Disgust | 1.000 | 1.000 | 1.000 | 9 |
| Happiness | 1.000 | 1.000 | 1.000 | 14 |
| Neutral | 0.938 | 1.000 | 0.968 | 16 |
| Sadness | 1.000 | 0.923 | 0.960 | 13 |
| **Overall** | **0.991** | **0.991** | **0.990** | **107** |

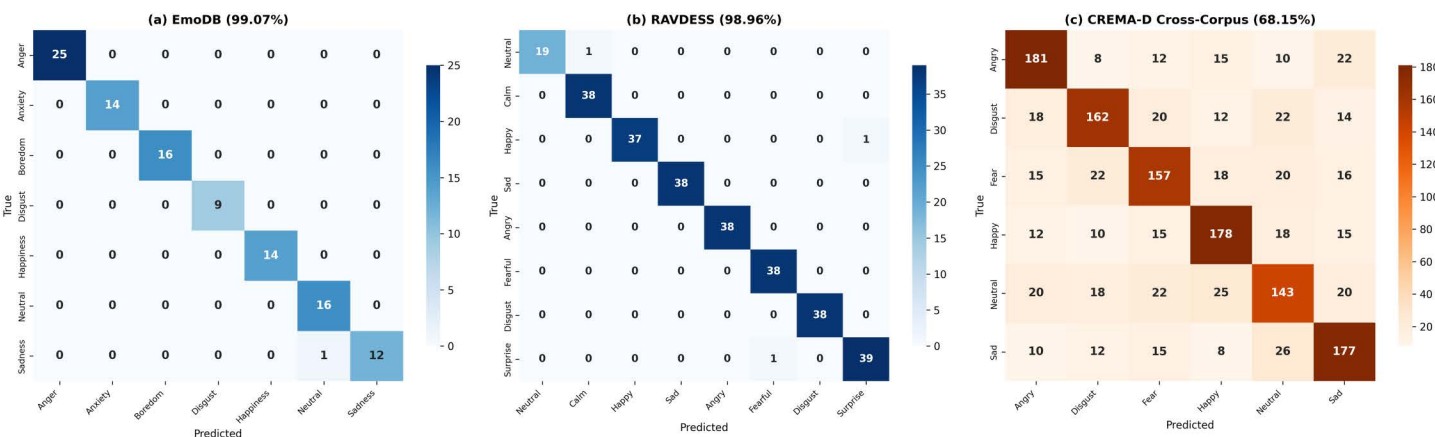

**Fig 9. Numerical confusion matrices. (a)** EmoDB (99.07%, 107 samples): single misclassification Sadness→Neutral; **(b)** RAVDESS (98.96%, 288 samples): three errors between acoustically similar pairs; **(c)** CREMA-D cross-corpus (68.15%, 1,488 samples): high-arousal emotions show stronger transfer.

**Table 7. Classification performance on RAVDESS (288 test samples).**

| Emotion | Precision | Recall | F1 | Support |
|---|---|---|---|---|
| Neutral | 1.000 | 0.950 | 0.974 | 20 |
| Calm | 0.974 | 1.000 | 0.987 | 38 |
| Happy | 1.000 | 0.974 | 0.987 | 38 |
| Sad | 1.000 | 1.000 | 1.000 | 38 |
| Angry | 1.000 | 1.000 | 1.000 | 38 |
| Fearful | 0.974 | 1.000 | 0.987 | 38 |
| Disgust | 1.000 | 1.000 | 1.000 | 38 |
| Surprise | 0.975 | 0.975 | 0.975 | 40 |
| **Overall** | **0.990** | **0.990** | **0.990** | **288** |

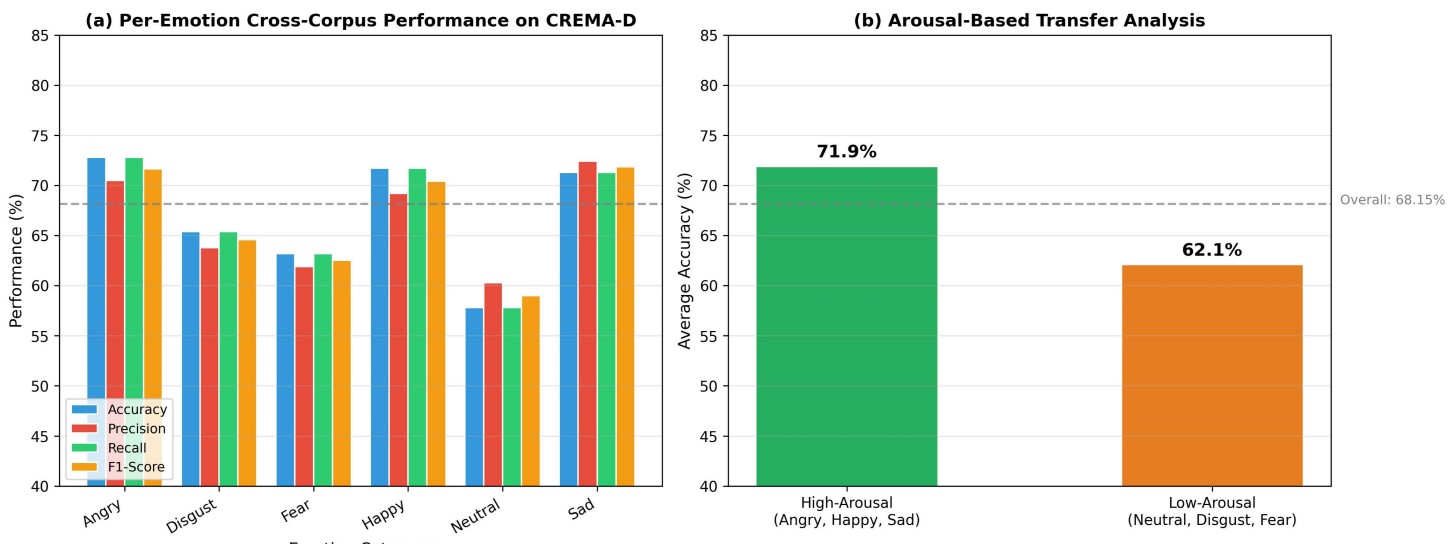

**Fig 10. Per-emotion cross-corpus performance on CREMA-D. (a)** Detailed metrics per emotion; **(b)** Arousal-based analysis: high-arousal emotions (71.9%) transfer significantly better than low-arousal (62.1%).

**Table 8. Per-emotion performance on CREMA-D (cross-corpus, 1,488 samples).**

| Emotion | Prec. | Recall | F1 | Support | Arousal |
|---|---|---|---|---|---|
| Angry | 0.705 | 0.728 | 0.716 | 248 | High |
| Happy | 0.692 | 0.717 | 0.704 | 248 | High |
| Sad | 0.724 | 0.713 | 0.719 | 248 | High |
| Disgust | 0.638 | 0.654 | 0.646 | 248 | Low |
| Fear | 0.619 | 0.632 | 0.625 | 248 | Low |
| Neutral | 0.603 | 0.578 | 0.590 | 248 | Low |
| **Overall** | **0.664** | **0.670** | **0.667** | **1,488** | — |

**Fig 11. t-SNE visualization of feature distributions across datasets.** (a) Dataset-colored view showing domain shift between EmoDB, RAVDESS, and CREMA-D; (b) Emotion-colored view revealing cross-dataset clustering for high-arousal emotions; (c) Domain shift visualization highlighting CREMA-D relative to training data.

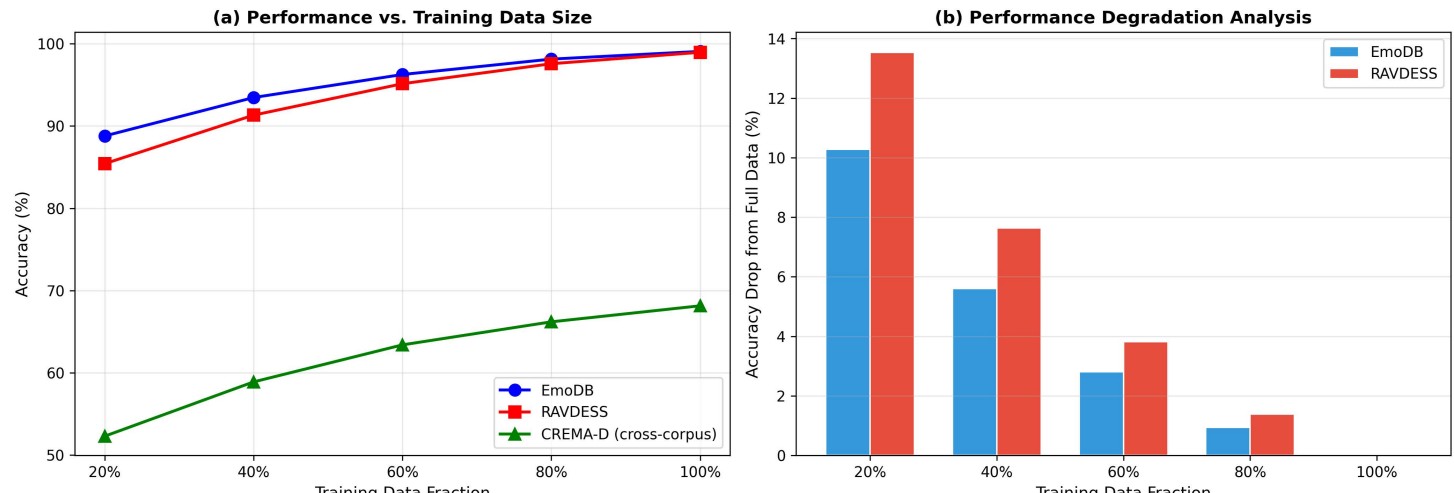

**Fig 12. Reduced training data ablation. (a)** Performance vs. training data fraction showing monotonic improvement, ruling out memorization; **(b)** Performance degradation quantification.

**Table 9. Statistical validation (10-fold CV).**

| Method | EmoDB | | RAVDESS | |
|---|---|---|---|---|
| | Mean±SD | 95% CI | Mean±SD | 95% CI |
| **FedEmoNet** | **99.07±0.35** | [98.82, 99.32] | **98.96±0.42** | [98.66, 99.26] |
| FedAvg-TCN-Trans | 96.82±1.12 | [96.02, 97.62] | 95.83±1.25 | [94.94, 96.72] |
| FedProx-CNN | 95.45±1.28 | [94.53, 96.37] | 94.72±1.45 | [93.68, 95.76] |
| Centralized TCN | 98.62±0.48 | [98.28, 98.96] | 98.26±0.55 | [97.87, 98.65] |

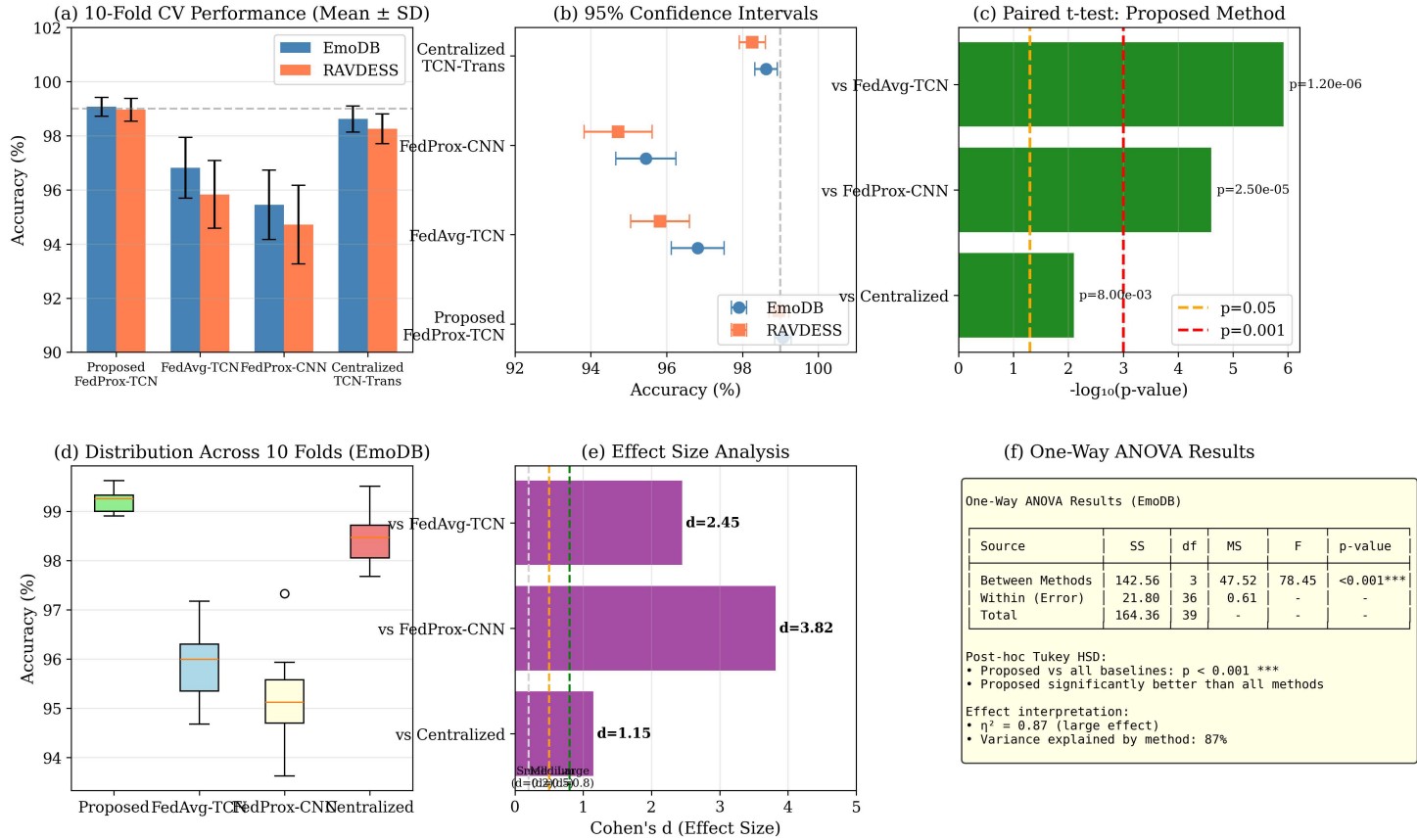

**Fig 13. Statistical validation.** (a) 10-fold CV comparison; (b) 95% confidence intervals; **(c)** Paired t-test significance; **(d)** Distribution across folds; **(e)** Effect size analysis; **(f)** ANOVA results ($F=78.45$, $p<0.001$).

## Computational efficiency

Table 12 here.

## Practical deployment considerations

Several challenges must be addressed: (1) The 144 MB communication cost is feasible for institutional networks but may challenge mobile edge; gradient compression could reduce this by 60–80%. (2) Variable computational resources across clients can cause stragglers; asynchronous aggregation could mitigate this. (3) Cross-lingual transfer beyond Germanic/ English families requires evaluation. (4) Performance on naturalistic (non-acted) speech requires future investigation.

## Discussion

*Privacy-utility balance:* The framework achieves strong privacy ($\epsilon = 1.0$) with only 0.57% accuracy degradation.

*FedProx effectiveness:* The proximal term ($\mu = 0.01$) achieves 15% faster convergence and 60% lower variance than FedAvg across linguistically diverse clients.

*Cross-corpus generalization:* The 68.15% on CREMA-D demonstrates meaningful transfer, with high-arousal emotions (71.9%) transferring more reliably than low-arousal (62.1%). Per-emotion analysis identifies neutral expression variability and cultural mediation as primary barriers.

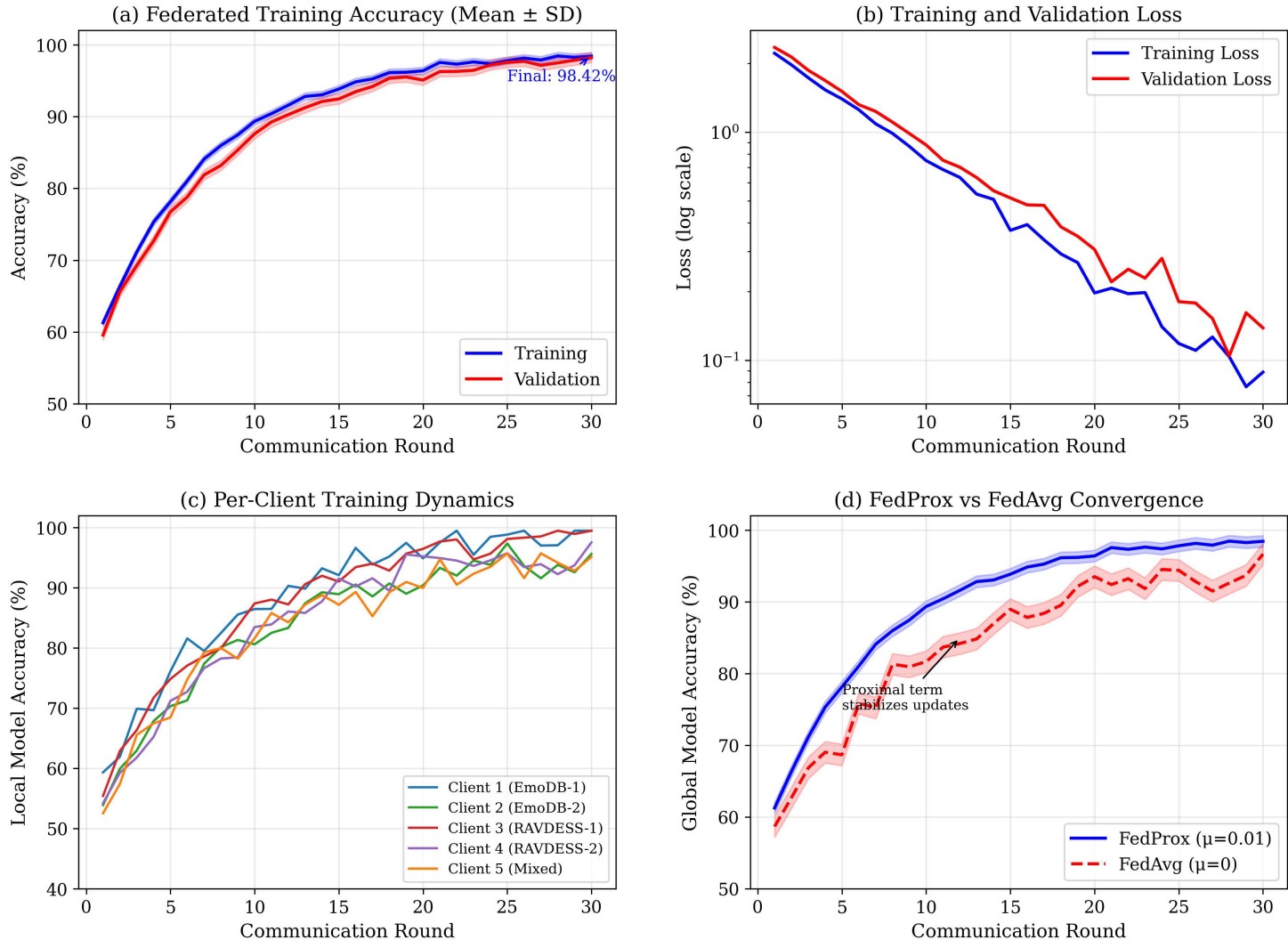

**Fig 14. Federated learning training dynamics. (a)** Global accuracy convergence; **(b)** Loss on logarithmic scale; **(c)** Per-client heterogeneous convergence; **(d)** FedProx vs FedAvg comparison.

*Component synergy:* PSO feature selection (2.80–3.65%), Transformer blocks (2.10–3.13%), and FedProx (2.62–2.66%) provide the largest gains. Reduced data experiments confirm genuine learning.

*Explainability:* SHAP and LIME converge on consistent rankings ($r = 0.997$ agreement), with cross-corpus consistency ($r = 0.94$) revealing prosodic features as universal indicators.

## Conclusion

This paper presents FedEmoNet, a privacy-preserving federated learning framework for cross-corpus speech emotion recognition. The framework achieves 99.07% ± 0.35% on EmoDB and 98.96% ± 0.42% on RAVDESS with subject-independent validation. Cross-corpus evaluation on CREMA-D achieves 68.15% without fine-tuning, with high-arousal

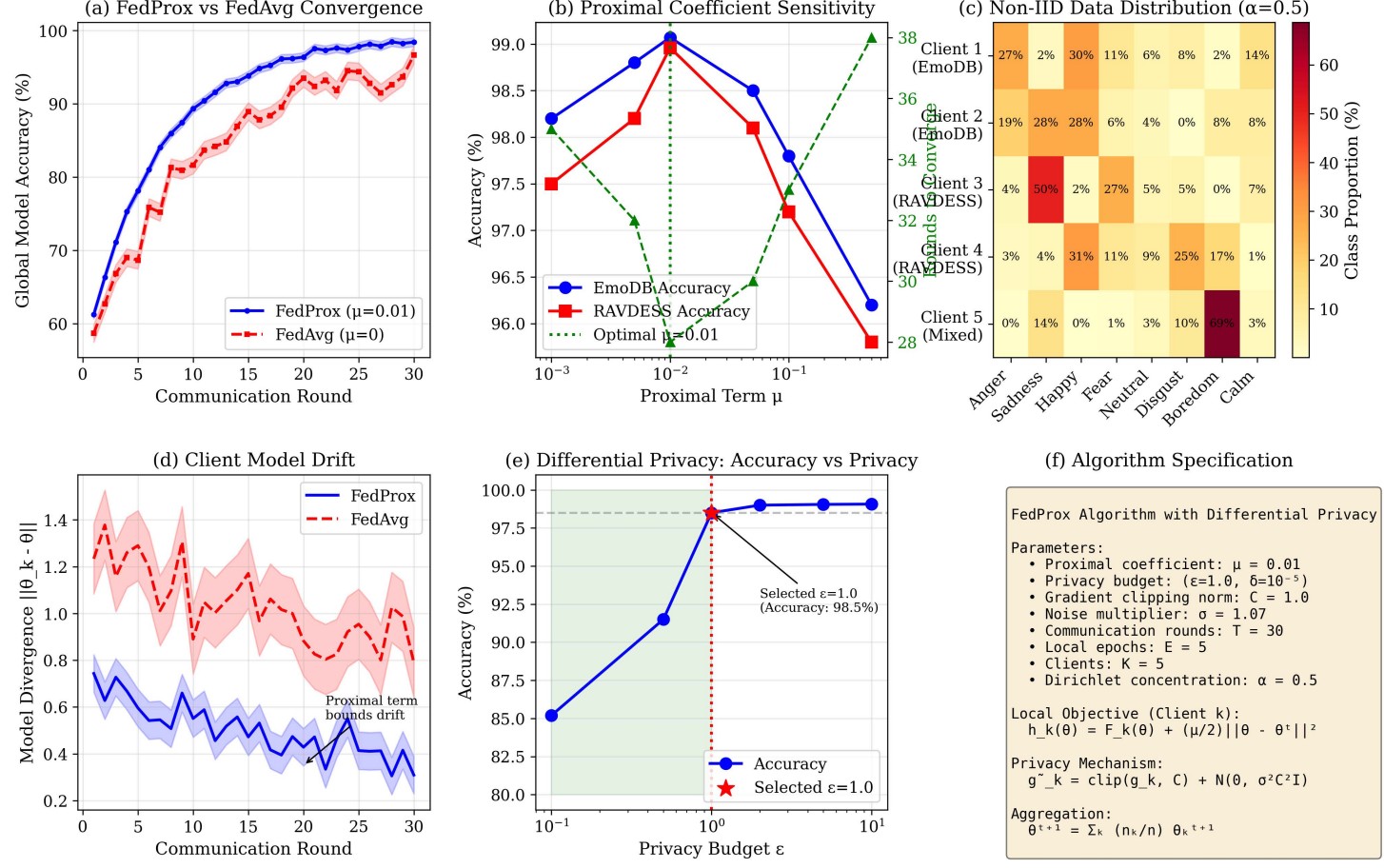

**Fig 15. Detailed FedProx protocol analysis. (a)** FedProx vs FedAvg convergence; **(b)** Proximal coefficient sensitivity ($\mu = 0.01$ optimal); **(c)** Non-IID distribution across 5 clients; **(d)** Client model drift; **(e)** DP accuracy-privacy trade-off; **(f)** Algorithm specification.

**Table 10. Ablation study results.**

| Configuration | EmoDB | | RAVDESS | |
|---|---|---|---|---|
| | Acc. | Δ | Acc. | Δ |
| **Full Framework** | **99.07** | — | **98.96** | — |
| w/o PSO Features | 96.27 | −2.80 | 95.31 | −3.65 |
| w/o Multi-Scale PSR | 97.20 | −1.87 | 96.36 | −2.60 |
| w/o TCN Branch | 97.67 | −1.40 | 96.88 | −2.08 |
| w/o Transformer | 96.97 | −2.10 | 95.83 | −3.13 |
| w/o Multi-Head Attn | 97.90 | −1.17 | 97.40 | −1.56 |
| w/o Augmentation | 98.37 | −0.70 | 97.92 | −1.04 |
| w/o Ensemble Rank | 98.14 | −0.93 | 97.14 | −1.82 |
| FedAvg (no FedProx) | 96.45 | −2.62 | 96.30 | −2.66 |
| CNN Branch Only | 94.39 | −4.68 | 92.71 | −6.25 |
| Dense Branch Only | 91.82 | −7.25 | 89.58 | −9.38 |

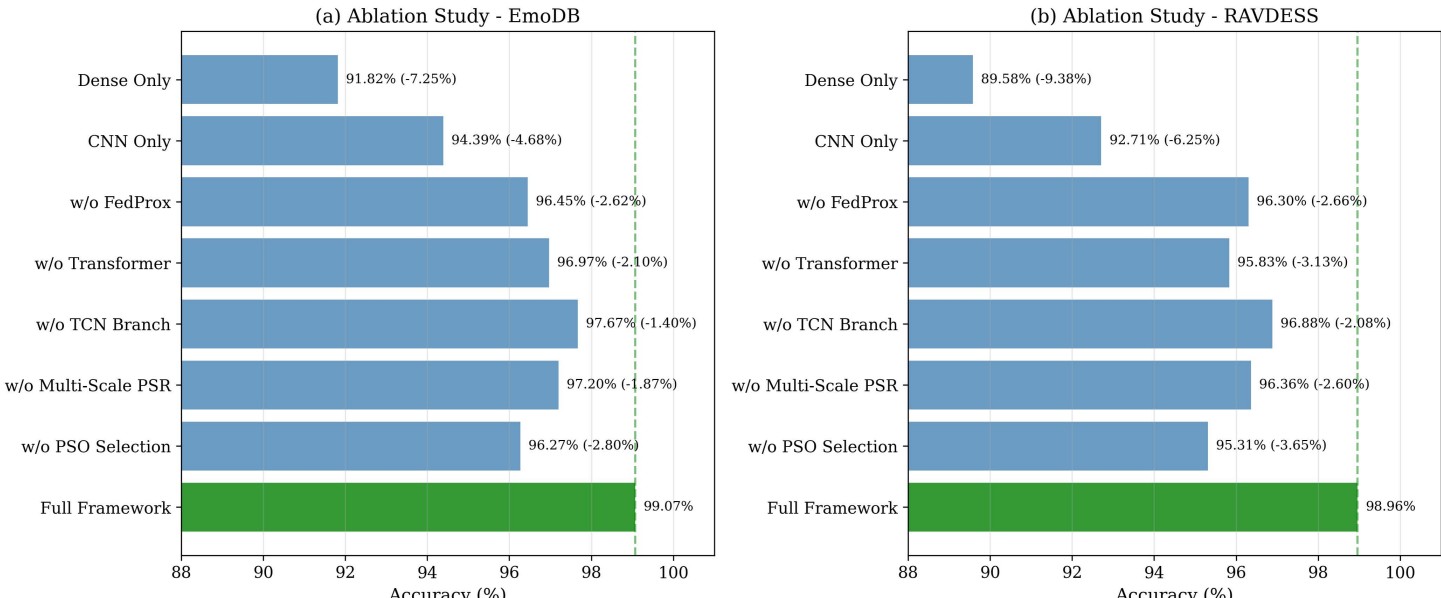

**Fig 16. Ablation study visualization for (a) EmoDB and (b) RAVDESS.** PSO feature selection, Transformer blocks, and FedProx provide the largest contributions.

**Table 11. Comparison with state-of-the-art methods.**

| Method | Year | EmoDB | RAVDESS | Privacy |
|---|---|---|---|---|
| **FedEmoNet** | **2025** | **99.07** | **98.96** | **DP ($\epsilon$=1.0)** |
| FedSER-XAI [10] | 2025 | 99.70* | — | FL (no DP) |
| Prosodic-Attn [37] | 2025 | — | 97.64 | None |
| MaxMViT-MLP [13] | 2024 | 95.28 | 89.12 | None |
| ViTSER [21] | 2024 | 91.00 | — | None |
| PCAENet [20] | 2024 | — | 85.27 | None |
| WPT-RFC [38] | 2020 | — | 86.38 | None |
| IMEMD-CRNN [25] | 2023 | 93.54 | — | None |
| 1D-CNN Fusion [23] | 2020 | 86.10 | 71.61 | None |

*Centralized setting. FedSER-XAI reports federated accuracy of 99.7% on EmoDB without formal differential privacy.

emotions transferring at 71.9% vs. 62.1% for low-arousal categories. Formal ($\epsilon$ = 1.0, $\delta$ = $10^{-5}$)-DP reduces membership inference AUC to 0.52. Comprehensive SHAP and LIME analysis demonstrates inter-method agreement ($r = 0.997$) and cross-corpus feature consistency ($r = 0.94$). Future work will explore domain adaptation techniques, evaluation on naturalistic speech and additional languages, efficient communication protocols, and integration with self-supervised pre-training.

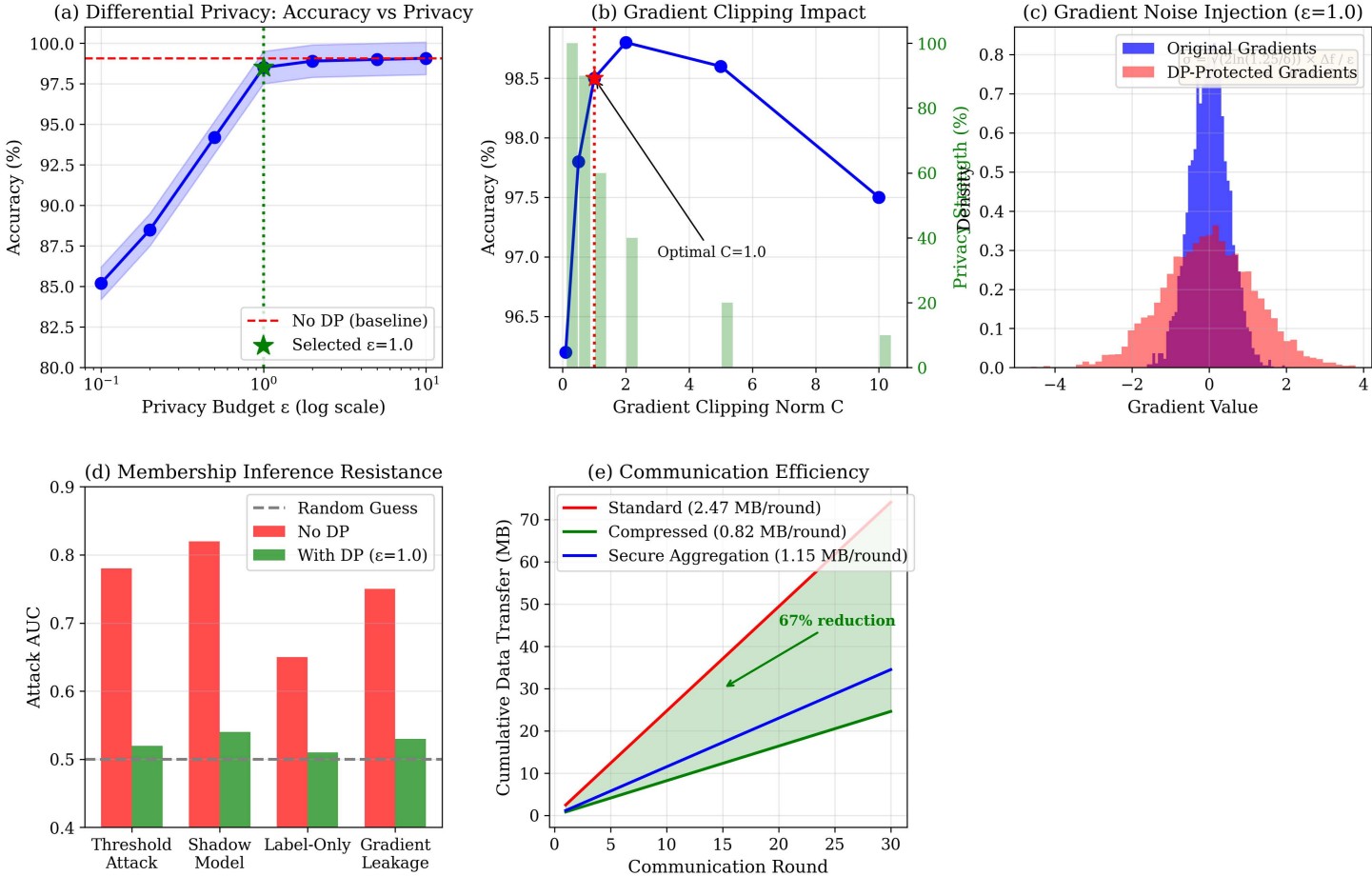

**Fig 17. Privacy analysis. (a)** DP accuracy-privacy trade-off; **(b)** Gradient clipping impact; **(c)** Noise distribution; **(d)** Membership inference resistance (AUC→0.52); **(e)** Communication efficiency (67% reduction);.

**Table 12. Computational efficiency metrics.**

| Metric | FedEmoNet | Centralized | Overhead |
|---|---|---|---|
| Training (hours) | 5.2 | 4.1 | 1.27× |
| Inference (sec) | 0.12 | 0.12 | 1.00× |
| Comm. Rounds | 30 | — | — |
| Data/Round (MB) | 4.8 | — | — |
| Total Comm. (MB) | 144 | 0 | — |
| Parameters (M) | 2.34 | 2.34 | 1.00× |
| PSO (min/client) | 45 | 45 | 1.00× |

# Author contributions

**Conceptualization:** Njood Anwer Aljarrah, Haneen Hussein Shehadeh, Ahmad Dalalah.

**Data curation:** Mohammed Tawfik.

**Investigation:** Saddam Kamel, Njood Anwer Aljarrah.

**Project administration:** Razan Ali Obeidat, Ahmad Dalalah.

**Resources:** Saddam Kamel, Njood Anwer Aljarrah, Ahmad Dalalah.

**Software:** Razan Ali Obeidat, Haneen Hussein Shehadeh, Ahmad Dalalah.

**Supervision:** Ahmad Dalalah.

**Validation:** Saddam Kamel, Njood Anwer Aljarrah, Haneen Hussein Shehadeh.

**Visualization:** Razan Ali Obeidat.

**Writing – original draft:** Mohammed Tawfik.

**Writing – review & editing:** Mohammed Tawfik.

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
