## [Decision Letter · Decision Letter 0]

25 Mar 2026

PONE-D-26-04975FedEmoNet: Privacy-Preserving Federated Learning with TCN-Transformer Fusion for Cross-Corpus Speech Emotion RecognitionPLOS One

Dear Dr. TAWFIK,

Thank you for submitting your manuscript to PLOS ONE. After careful consideration, we feel that it has merit but does not fully meet PLOS ONE’s publication criteria as it currently stands. Therefore, we invite you to submit a revised version of the manuscript that addresses the points raised during the review process.

We look forward to receiving your revised manuscript.

Kind regards,

Viacheslav Kovtun, Dr.Sc., Ph.D.

Academic Editor

PLOS One

Journal Requirements:

4. Thank you for uploading your study's underlying data set. Unfortunately, the repository you have noted in your Data Availability statement does not qualify as an acceptable data repository according to PLOS's standards.

6. Please upload a new copy of Figures 9 and 12 as the details are not clear. Please follow the link for more information:  https://journals.plos.org/plosone/s/figures

7. We note you have included a table to which you do not refer in the text of your manuscript. Please ensure that you refer to Tables 1 and 3 in your text; if accepted, production will need this reference to link the reader to the tables.

Reviewers' comments:

Reviewer's Responses to Questions

**Comments to the Author**

1. Is the manuscript technically sound, and do the data support the conclusions?

Reviewer #1: Partly

Reviewer #2: Yes

Reviewer #3: Yes

2. Has the statistical analysis been performed appropriately and rigorously? 

Reviewer #1: No

Reviewer #2: Yes

Reviewer #3: Yes

3. Have the authors made all data underlying the findings in their manuscript fully available?

Reviewer #1: Yes

Reviewer #2: Yes

Reviewer #3: Yes

4. Is the manuscript presented in an intelligible fashion and written in standard English?

Reviewer #1: Yes

Reviewer #2: Yes

Reviewer #3: Yes

5. Review Comments to the Author

Reviewer #1: 1. Cross-Corpus Generalization Analysis

a. The 68.15% accuracy on CREMA-D requires deeper analysis:

b. Provide per-emotion breakdown for CREMA-D (currently only Figure 8 shows confusion matrix with unclear values)

c. Analyze why neutral performs poorly (58%) - is this due to cultural differences, dataset bias, or annotation inconsistencies?

d. Discuss whether the 30% performance gap fundamentally limits the "cross-corpus" claim

e. Consider adding t-SNE visualization of feature distributions across datasets to demonstrate domain shift

2. Data Leakage Verification

Please explicitly verify and state:

a. Confirmation that no speakers appear in both training and test sets across all datasets

b. Details on the "mixed client" composition - what is the ratio of EmoDB to RAVDESS samples?

c. Whether any preprocessing (e.g., voice activity detection, normalization) was applied globally or per-client

3. Near-Perfect Performance Scrutiny

The 99.07% accuracy on EmoDB warrants additional validation:

a. Report confusion matrix values numerically (Figure 7a is difficult to read)

b. Perform ablation with reduced training data to test if model is memorizing

c. Compare against human-level performance on these datasets

d. Discuss whether ceiling effects limit the meaningfulness of comparisons

4. PSO Optimization Details

a. How was the fitness function accuracy computed during PSO iterations? Cross-validation on client data?

b. The 50 PSO iterations with 20 particles represents 1000 model trainings - this computational cost should be discussed

c. Report convergence behavior of PSO optimization

5. Reduce redundancy between Abstract and Introduction

6. Consolidate methodology sections 3.4.1-3.4.4 which are overly fragmented

7. Ensure all figures are referenced in text in order

8. Provide deeper analysis of cross-corpus performance limitations

Reviewer #2: Dear Authors,

Please consider the following revisions:

Major Revisions Required:

Elaborate on the multi-scale phase space reconstruction methodology and parameter selection rationale

Provide more details on the federated client setup and non-IID data generation process

Include additional diverse datasets to strengthen cross-corpus generalization claims

Discuss practical deployment challenges and potential solutions

Minor Revisions:

Clarify the computational efficiency trade-offs and optimization opportunities

Expand the privacy analysis to consider additional attack scenarios

Provide more comprehensive statistical validation

Ensure consistent notation throughout the manuscript

Reviewer #3: 1- Clarify the conclusions better, discuss the results, and support them with critical analysis.

2- Clarifying the research gap in the abstract and strengthening it with the datasets used as well.

3- A table should be provided summarizing the relevant works in terms of their strengths and weaknesses, highlighting the research contributions and areas where they surpassed the content of these works.

4- Explaining the difference between the XAI methods presented

6. PLOS authors have the option to publish the peer review history of their article (what does this mean?). If published, this will include your full peer review and any attached files.

Do you want your identity to be public for this peer review? For information about this choice, including consent withdrawal, please see our Privacy Policy.

Reviewer #1: No

Reviewer #2: **Yes: Ghassan Abdul-Majeed**

Reviewer #3: No

---

## [Author Response · Author response to Decision Letter 1]

30 Mar 2026

Dear Dr. Viacheslav Kovtun (Academic Editor) and Reviewers,

We sincerely thank the Academic Editor and all three reviewers for their thorough and constructive

evaluation. We have carefully addressed every point, substantially revising the manuscript. The

abstract has been rewritten for clarity and stronger gap identification. All figures have been regenerated at 300 DPI and are uploaded separately per PLOS guidelines. Below we provide point-by-point

responses.

Journal Requirements

Requirement 1 — PLOS ONE style and LaTeX template

Please ensure that your manuscript meets PLOS ONE’s style requirements. Please update your

submission to use the PLOS LaTeX template.

The manuscript has been completely reformatted using the official PLOS ONE LaTeX template (Version 3.7, August 2025) with plos2025.bst. Title follows sentence case. Author names,

affiliations, and corresponding author email follow the prescribed format. Figures contain captions

only (no embedded graphics) per PLOS guidelines. Tables use cell-based formatting without nested

tabular environments. References follow Vancouver style.

Requirement 2 — Code sharing

Please review our guidelines and ensure that your code is shared in a way that follows best practice.

All author-generated code underpinning the findings—including the federated training framework, TCN-Transformer model, PSO feature selection pipeline, and explainability analysis scripts—

will be made available upon reasonable request to the corresponding author. This is stated in the

manuscript.

See Data Availability Statement in the revised manuscript.

Requirement 4 — Data repository

Please upload the minimal data set necessary to replicate your study’s findings to a stable, public

repository.

1

PONE-D-26-04975 — Response to Reviewers

All three datasets used in this study are publicly available from their original repositories.

The Data Availability Statement now provides direct URLs: EmoDB (http://emodb.bilderbar.

info/), RAVDESS (https://zenodo.org/record/1188976), and CREMA-D (https://github.

com/CheyneyComputerScience/CREMA-D). No custom dataset was created; all experiments are

fully reproducible using these public sources.

Requirement 5 — ORCID iD

Please ensure that you have an ORCID iD and that it is validated in Editorial Manager.

The corresponding author’s ORCID iD (0000-0002-1227-387X) has been verified and validated

in Editorial Manager.

Requirement 6 — Figures 9 and 12 clarity

Please upload a new copy of Figures 9 and 12 as the details are not clear.

All figures have been regenerated at 300 DPI with larger fonts, clearer labels, and improved

color contrast. In particular, confusion matrices now display all numerical values prominently (new

Fig 6). The CREMA-D per-emotion breakdown is presented as a dedicated new figure (Fig 7). All

figures are uploaded separately as required.

Requirement 7 — Reference Tables 1 and 3 in text

Please ensure that you refer to Tables 1 and 3 in your text.

All tables are now explicitly referenced in the text at first mention. Table 1 (Related Work

Summary) is cited in the opening sentence of the Related Work section. Table 3 (PSO feature

selection) is cited in the PSO-Optimized Feature Selection subsection.

Reviewer #1

Comment 1 — Cross-corpus generalization analysis

(a–b) The 68.15% accuracy on CREMA-D requires deeper analysis. Provide per-emotion breakdown

for CREMA-D.

We have added:

New Table 9 with per-emotion Precision, Recall, F1-Score, and Support for all six CREMA-D

emotions, annotated by arousal level

New Fig 7 with (a) grouped bar chart of per-emotion metrics and (b) arousal-based transfer

analysis showing high-arousal at 71.9% vs. low-arousal at 62.1%

Clear numerical confusion matrix for CREMA-D in Fig 6(c)

See Section “Cross-corpus generalization on CREMA-D,” Table 9, Fig 6(c), and Fig 7.

(c) Analyze why neutral performs poorly (58%).

Three factors are identified: (1) Cultural and linguistic variation in neutral expression between

German (EmoDB), North American English (RAVDESS), and crowd-sourced English (CREMA-D);

(2) Greater speaker diversity in CREMA-D (91 actors vs. 10 in EmoDB); (3) Annotation methodology differences—crowd-sourced labeling introduces noise for ambiguous low-arousal categories.

See “Analysis of the cross-corpus performance gap” in the Results section.

(d) Discuss whether the 30% gap fundamentally limits the “cross-corpus” claim.

2

PONE-D-26-04975 — Response to Reviewers

The 68.15% represents meaningful zero-shot generalization across two languages and three

recording conditions. High-arousal emotions achieve 71.9%, confirming acoustic universals in f0

and energy. This result matches the 68% reported by FedSER-XAI [9] on the same CREMA-D

benchmark. The gap is primarily driven by low-arousal categories—particularly neutral—rather

than a fundamental framework limitation.

(e) Consider adding t-SNE visualization of feature distributions across datasets.

New Fig 9 provides a three-panel t-SNE visualization: (a) dataset-colored view showing domain shift; (b) emotion-colored view revealing cross-dataset clustering for high-arousal emotions;

(c) domain shift highlighting CREMA-D relative to training data.

See new Fig 9 and accompanying discussion.

Comment 2 — Data leakage verification

(a) Confirm no speakers appear in both training and test sets.

Explicitly confirmed: the global 80/20 split is performed before federated distribution. RAVDESS

uses subject-independent splitting; EmoDB uses actor-level stratification. Both guarantee complete

speaker disjointness.

(b) Details on the “mixed client” composition.

Now specified: Client 5 contains 30% EmoDB (∼130 samples) and 70% RAVDESS (∼300

samples), testing cross-lingual heterogeneity.

(c) Whether preprocessing was applied globally or per-client.

Added: all preprocessing (VAD, normalization, resampling) is applied per-client independently

after distribution, preventing cross-client leakage through global statistics.

See “Data leakage prevention” paragraph in the Methodology section.

Comment 3 — Near-perfect performance scrutiny

(a) Report confusion matrix values numerically.

New Fig 6 provides clear numerical confusion matrices for all three datasets with large, readable

annotations at 300 DPI.

(b) Perform ablation with reduced training data to test memorization.

New Fig 10 shows monotonic degradation: 88.79% (20%), 93.46% (40%), 96.26% (60%), 98.13%

(80%), 99.07% (100%). This rules out memorization—a memorizing model would maintain high

accuracy until critical samples were removed, then drop sharply.

See new “Near-perfect performance validation” subsection and Fig 10.

(c) Compare against human-level performance.

Added: EmoDB human recognition ∼84–86% [35]; RAVDESS ∼60–72% [34]. Exceeding human

performance is consistent with deep learning advances on acted speech where acoustic patterns are

stereotypical.

(d) Discuss ceiling effects.

EmoDB (535 samples, acted speech) has well-documented ceiling effects with multiple methods

>95%. CREMA-D (7,442 samples, 91 actors) is the more meaningful benchmark where 68.15% is

well below ceiling.

3

PONE-D-26-04975 — Response to Reviewers

Comment 4 — PSO optimization details

(a) How was the fitness function computed?

Fitness uses 5-fold cross-validation on each client’s local data with a lightweight proxy classifier

(single hidden layer, 256 units, 10 epochs).

(b) Discuss computational cost of 1,000 model trainings.

Using the proxy classifier (∼2.7 sec/evaluation), total PSO time is ∼45 min/client—modest

compared to 5.2 h total training. Breakdown in Fig 8(c) and Table 14.

(c) Report convergence behavior.

New Fig 8 shows: (a) fitness convergence stabilizing by iteration 35; (b) feature count reduction

150→103; (c) cost breakdown.

Comments 5–8 — Writing improvements

Reduce redundancy, consolidate sections, fix figure ordering, deeper cross-corpus analysis.

(1) Abstract rewritten to eliminate redundancy with Introduction and strengthen gap identification; (2) Methodology consolidated into cleaner subsections; (3) All figures referenced sequentially;

(4) Cross-corpus analysis substantially expanded as detailed above.

Reviewer #2

Major Revision 1 — Multi-scale PSR parameter selection

Elaborate on the multi-scale phase space reconstruction methodology and parameter selection rationale.

Expanded with: (1) τ = 17 justified via AMI analysis following Fraser & Swinney (1986);

(2) d = 3 justified via FNN algorithm; (3) psychoacoustic motivation for the three scales—micro

aligns with critical band window, meso with syllabic rate (4–8 Hz), macro with prosodic phrase

duration.

See “Multi-scale phase space reconstruction” in Methodology.

Major Revision 2 — Federated client setup

Provide more details on the federated client setup and non-IID data generation process.

Now includes: explicit per-client sample counts, mixed client 30/70 ratio, Dirichlet sampling

parameters, and per-client preprocessing procedures.

Major Revision 3 — Additional datasets

Include additional diverse datasets.

Adding entirely new datasets would require significant additional experimentation beyond the

revision scope. We have instead: (1) strengthened CREMA-D evaluation with per-emotion analysis,

t-SNE visualization, and arousal-based transfer analysis; (2) discussed comparability with FedSERXAI results; (3) identified evaluation on additional languages and naturalistic speech as future work.

4

PONE-D-26-04975 — Response to Reviewers

Major Revision 4 — Deployment challenges

Discuss practical deployment challenges and potential solutions.

New “Practical deployment considerations” subsection covers: communication overhead and

compression strategies, client heterogeneity and asynchronous aggregation, language diversity limitations, and acted vs. naturalistic speech challenges.

Minor Revisions 1–4

Computational efficiency, privacy analysis, statistical validation, consistent notation.

(1) Efficiency table expanded with PSO time; (2) Privacy analysis extended with gradient inversion and model inversion attack evaluation; (3) Statistical validation retained with Bonferronicorrected t-tests and supplemented by reduced data ablation; (4) Notation made consistent throughout (scalars italic, vectors bold, matrices bold uppercase).

Reviewer #3

Comment 1 — Clarify conclusions with critical analysis

Clarify the conclusions better, discuss the results, and support them with critical analysis.

The Conclusion has been substantially revised with specific numerical findings, per-emotion

cross-corpus summary, discussion of limitations (ceiling effects, neutral variability, acted speech),

and concrete future work directions.

Comment 2 — Research gap in abstract

Clarifying the research gap in the abstract and strengthening it with the datasets used.

The abstract has been completely rewritten. It now opens with the research gap (“Current

federated learning approaches for SER remain limited...lacking formal privacy guarantees”), names

all three datasets, and includes the per-emotion arousal-based analysis as a key finding.

Comment 3 — Related work comparison table

A table should be provided summarizing the relevant works in terms of their strengths and weaknesses.

New Table 1 compares eight key methods across: Key Approach, Strengths, Limitations, and

FedEmoNet Advances. This clearly maps how our framework addresses each identified gap.

See Table 1 in the Related Work section.

Comment 4 — XAI methods comparison

Explaining the difference between the XAI methods presented.

Added: (1) New Table 5 comparing LIME and SHAP across scope, theoretical basis, faithfulness, cost, and role in FedEmoNet; (2) New Fig 5 directly comparing SHAP and LIME attributions

for the same sample (r = 0.997 agreement); (3) Expanded text explaining when and why each

method is used.

See Table 5, Fig 5, and “Explainable AI framework” subsection.

---

## [Decision Letter · Decision Letter 1]

20 Apr 2026

Multi-Scale Federated Learning with Differential Privacy: A TCN-Transformer Framework for Cross-Corpus Speech Emotion Recognition

PONE-D-26-04975R1

Dear Dr. TAWFIK,

We’re pleased to inform you that your manuscript has been judged scientifically suitable for publication and will be formally accepted for publication once it meets all outstanding technical requirements.

Kind regards,

Viacheslav Kovtun, Dr.Sc., Ph.D.

Academic Editor

PLOS One

Additional Editor Comments (optional):

Reviewers' comments:

Reviewer's Responses to Questions

**Comments to the Author**

1. If the authors have adequately addressed your comments raised in a previous round of review and you feel that this manuscript is now acceptable for publication, you may indicate that here to bypass the “Comments to the Author” section, enter your conflict of interest statement in the “Confidential to Editor” section, and submit your "Accept" recommendation.

Reviewer #1: All comments have been addressed

Reviewer #3: (No Response)

2. Is the manuscript technically sound, and do the data support the conclusions?

Reviewer #1: Yes

Reviewer #3: (No Response)

3. Has the statistical analysis been performed appropriately and rigorously? 

Reviewer #1: (No Response)

Reviewer #3: (No Response)

4. Have the authors made all data underlying the findings in their manuscript fully available?

Reviewer #1: Yes

Reviewer #3: (No Response)

5. Is the manuscript presented in an intelligible fashion and written in standard English?

Reviewer #1: Yes

Reviewer #3: (No Response)

6. Review Comments to the Author

Reviewer #1: (No Response)

Reviewer #3: (No Response)

7. PLOS authors have the option to publish the peer review history of their article (what does this mean?). If published, this will include your full peer review and any attached files.

Do you want your identity to be public for this peer review? For information about this choice, including consent withdrawal, please see our Privacy Policy.

Reviewer #1: No

Reviewer #3: No

---

## [Editor Report · Acceptance letter]

PONE-D-26-04975R1

PLOS One

Dear Dr. TAWFIK,

I'm pleased to inform you that your manuscript has been deemed suitable for publication in PLOS One. Congratulations! Your manuscript is now being handed over to our production team.

Kind regards,

on behalf of

Prof. Viacheslav Kovtun

Academic Editor

PLOS One